# 'Well London' and the benefits of participation: results of a qualitative study nested in a cluster randomised trial

Jane Derges,[1] Angela Clow,[2] Rebecca Lynch,[1] Sumeet Jain,[1] Gemma Phillips,[3] Mark Petticrew,[4] Adrian Renton,[3] Alizon Draper[1]

▶ Prepublication history and additional material is available. To view please visit the journal online (http://dx.doi.org/10.1136/bmjopen-2013-003596).

[1]School of Life Sciences, University of Westminster, London, UK
[2]Department of Psychology, University of Westminster, London, UK
[3]Institute for Health and Human Development, University of East London, London, UK
[4]Department of Public Health, London School of Hygiene and Tropical Medicine, London, UK

Correspondence to
Dr Jane Derges;
jane.derges@bristol.ac.uk

## ABSTRACT

**Background:** *Well London* is a multicomponent community engagement and coproduction programme designed to improve the health of Londoners living in socioeconomically deprived neighbourhoods. To evaluate outcomes of the *Well London* interventions, a cluster randomised trial (CRT) was conducted that included a longitudinal qualitative component, which is reported here. The aim is to explore in depth the nature of the benefits to residents and the processes by which these were achieved.

**Methods:** The 1-year longitudinal qualitative study was nested within the CRT. Purposive sampling was used to select three intervention neighbourhoods in London and 61 individuals within these neighbourhoods. The interventions comprised activities focused on: healthy eating, physical exercise and mental health and well-being. Interviews were conducted at the inception and following completion of the *Well London* interventions to establish both if and how they had participated. Transcripts of the interviews were coded and analysed using Nvivo.

**Results:** Positive benefits relating to the formal outcomes of the CRT were reported, but only among those who participated in project activities. The extent of benefits experienced was influenced by factors relating to the physical and social characteristics of each neighbourhood. The highest levels of change occurred in the presence of: (1) social cohesion, not only pre-existing but also as facilitated by *Well London* activities; (2) personal and collective agency; (3) involvement and support of external organisations. Where the physical and social environment remained unchanged, there was less participation and fewer benefits.

**Conclusions:** These findings show interaction between participation, well-being and agency, social interactions and cohesion and that this modulated any benefits described. Pathways to change were thus complex and variable, but personal well-being and local social cohesion emerged as important mediators of change.

## Strengths and limitations of this study

- Uses participants' perspectives to identify why and how people participate in community health programmes.
- Focuses on agency and its relationship to 'well-being' within an urban marginalised population.
- Highlights the importance of social cohesion to 'well-being' and subsequently, participation in community health programmes.
- Further exploration of non-participation is required for future study.
- An ethnographic focus would contribute both methodologically and to analysis.

the UK remains a key public health challenge,[1] [2] but while there is extensive evidence documenting the consequences and causes of health inequalities, less is known about the interplay between specific causal factors or what interventions are effective in reducing them.[3] There is growing recognition of the need to understand which interventions are effective and the processes or pathways by which effects are achieved.[4–7] This is particularly important for interventions that are 'complex'[8] [9] and in which local contextual factors modulate both the process of implementation and generation of outcomes.[10] The *Well London* programme is a complex intervention comprising multiple components and using a community engagement model. The term 'participation' is used here to highlight participants' agency in relation to choice and whether or not they decided to take part in the interventions, including as volunteers. The interventions comprised a series of activities based around healthy eating, physical activity and mental well-being. 'Well-being' is here defined as a eudemonic state in which the individual experiences positive attachment, a sense of meaningfulness and usefulness in life. This framework was used in each area but the delivery method varied according to local

## BACKGROUND
### Processes of change in community-based interventions

Improving the health and well-being of populations living in disadvantaged areas of

needs and priorities, as outlined in current theories concerning the design and evaluation of complex interventions.[11] [12] Further details are obtainable from the *Well London* website.[13]

As Draper *et al* and others note there is a need for rigorous evaluations that explore the causal pathways by which community participation influences health outcomes.[14–16] Popay[17] and Wallerstein[18] have hypothesised a number of possible pathways, but these remain largely unexamined. The relationship between social context, individual agency and participation has also been neglected,[19] as well as exploration of the effects of interventions to address health inequalities[20] and the nature of personal agency and its relationship to social determinants of health framework.[21] While the social and environmental context in which people live and their ability to exercise individual agency in relation to decision-making about health is recognised as important, there are few qualitative studies that examine how these are interlinked and impact community engagement programmes.

The analysis presented here focuses on a qualitative study, which was embedded within the cluster randomised trial (CRT) of the *Well London* programme (for a full description of the trial protocol[22]). The primary aim of the qualitative study was to examine the causal pathways that generated any intervention effects from the perspectives of local residents, who were involved as strategic partners in *Well London*'s design and delivery.

## METHODS
### Study objectives
The objectives of the qualitative study were:
1. To identify how and why individuals participated in the *Well London* activities;
2. To explore the different project components that enabled people to improve their health practices and sense of 'well-being';
3. To identify factors in the social and physical environment that influenced attitudes to health.

### Study design
A longitudinal qualitative research element was included in the CRT in order (1) to address the complexity[23] of the intervention; (2) to try to identify specific factors which enable or obstruct individuals in leading healthy lives; (3) to understand subjective experience and the role of 'agency' in relation to participation.

The longitudinal qualitative study component was nested within the larger *Well London* CRT (for details of overall trial design[24]). It comprised a series of in depth interviews conducted over a period of 1 year, with interviews undertaken in two stages: first, at the implementation stage of the interventions and second, postintervention. Both participants and non-participants of *Well London* were interviewed to capture whether exposure to the interventions would lead to neighbourhood level improvements in health and health practices, or whether direct participation was

required. Limited observation of three selected *Well London* intervention neighbourhoods was also undertaken, to contextualise the interview data. Interviews were inductively examined in accordance with increasing calls for better capture of participants' views.[25]

### Selection of study neighbourhoods
Initial observation showed contextual variation between the 20 *Well London* intervention neighbourhoods in environment, demography and history as well as in intensity and range of community activities running parallel to (ie, not commissioned by) *Well London*. A critical case sampling approach was therefore used to select three neighbourhoods to be included in the qualitative study (chosen in consideration of the in depth nature of the study). This approach selects cases based on criteria that are seen to be particularly important for the research project: "if it happens there, it will happen everywhere" or "if it doesn't happen there it won't happen anywhere", "if that group is having problems we can be sure that every group is having problems".[26]

In selecting this approach it was necessary to identify what would make a critical case in relation to the objectives of the qualitative study. We therefore included neighbourhoods with low and high levels of community projects. The method of programme delivery within *Well London* also differed and the three neighbourhoods chosen reflect this by including neighbourhoods with high and low levels of pre-existing community activities beyond those provided by the *Well London* programme, and differences in the manner of their delivery.

The first of these neighbourhoods (Eastford[i]) had a wide range of community activities offered prior to *Well London*, and continues to offer many activities unconnected to *Well London*. The second neighbourhood (Hartfield) has a core group of volunteers instrumental in generating engagement in *Well London* activities and they live in an enclosed geographical space (a housing estate). The third neighbourhood (Mountside) had limited community activities prior to *Well London* and a population dispersed among a number of differing housing sites. The manner of *Well London* delivery in these neighbourhoods varied from highly proactive and involved members to a less cohesive and active method of delivery.

### Study population
Sixty-one individuals were recruited at the start of *Well London* delivery and comprised matched participants of the interventions, and non-participants (see table 1). Participants were purposively selected from within the interventions across the three study neighbourhoods and non-participants were selected through snowball sampling; these contacts were made by the researcher during visits to the neighbourhood. Selection for the

---

[i]Pseudonyms are used throughout for places and people.

qualitative study was based on providing theoretical insights rather than broader generalisations, as noted by Gardner and Chapple.[27] 'Participant' is here defined as a resident who received the *Well London* activities, and volunteered in their delivery.

For the postintervention interview, a total of 45 agreed to a second interview. Reasons given by the 16 who did not attend this second interview were as follows: moved out of the neighbourhood (2), refusal of a follow-up interview (3), no response elicited (9) and illness (2). New recruits were not sought as changes over the intervening period would not have been captured.

Ethnicity, age and length of time in the neighbourhood among the study population were mixed across all three neighbourhoods; each neighbourhood showed variation according to all these categories, most noticeably, ethnicity. It is beyond the scope of this paper to examine the effects of this in detail, other than to recognise this as a difference requiring further investigation.

## Data collection and analysis

Preintervention and postintervention interviews used the same topic guide and focused on participants' experiences of the *Well London* interventions and any reported changes to eating, exercise and mental health practices (see online supplementary file). Also participants and non-participants were asked for their views about the neighbourhood environment. Interviews were recorded and transcribed before being entered into Nvivo. Each transcript was checked for quality, coded and analysed using a framework based on Spencer, Ritchie and O'Conner's 'analytic hierarchy'[28]. This allowed systematic analysis of the large dataset but was flexible enough to allow refinements to the coding. Codes from the interviews were identified and grouped under categories generated from the interview topics. Data were analysed not only deductively from the primary outcome measures (changes to healthy eating, physical activity and mental well-being), but also inductively from emerging themes identified from the interviews. Observational data of the neighbourhoods were recorded in photographs and written notes and included the local geography, amenities and range of community facilities and activities that were available. A separate researcher was employed at each stage of the study; one researcher for stage 1 interviews, and the second researcher for stage 2. They each also conducted the observations simultaneous to the interviews.

## Interview quality assurance and ethics

Quality assurance procedures were undertaken to minimise researcher bias when coding the interviews by randomly selecting three interviews, which were then recoded by two independent researchers blind to the initial coding. The three interviews were compared to identify new codes and establish a degree of consensus in applying a particular code to similar text. Following initial telephone/email contact, written consent was obtained from every participant/non-participant. Each individual also received a short pamphlet describing the *Well London* project and an explanation of the qualitative component. Verbal explanation was also provided at the start of the interview process.

The results of the qualitative study are presented here by area. The reason for this is that the context and environment in which participants and non-participants were living and into which the *Well London* interventions and activities were introduced, has been shown to be a key factor in showing why individuals participated. Subsequently, data are presented by area, not theme.

**Table 1** Participant and non-participant profiles

| First round interviews | Age range | Ethnicity | Gender | *Well London* participation | Second round interviews |
|---|---|---|---|---|---|
| Hartfield | 16–25: 3<br>26–35: 8<br>36–45: 3<br>46–55: 5<br>56–65: 2<br>66–89: 1 | African 13, Indian 3, Bangladeshi 1, White British 3, European (Lithuania) 1 | Female: 16<br>Male: 5 | Participants: 13<br>Non-participant: 8<br>Total: 21 | Stage 2<br>Participant=11<br>Non-participant=8<br>Total: 20 |
| Eastford | 16–25: 4<br>26–35: 4<br>36–45: 2<br>46–55: 4<br>56–65: 3<br>66–75: 3 | Bangladeshi 5, Indian 1, Pakistani 1, Caribbean 2, Black British 1, African 3, Chinese 2, White British 3, Irish 2 | Female: 18<br>Male: 2 | Participants: 11<br>Non-participants: 9<br>Total: 20 | Stage 2<br>P=7<br>NP=8<br>Total: 15 |
| Mountside | 16–25: 5<br>26–35: 3<br>36–45: 4<br>46–55: 3<br>56–65: 3<br>66–75: 2 | White British 5, British Asian 5, Caribbean 4, European (Turkish) 3, African 2, Chinese 1 | Female: 11<br>Male: 9 | Participants: 10<br>Non-participant: 10<br>Total: 20 | Stage 2<br>P=6<br>NP=4<br>Total: 10 |

 

## RESULTS

Findings from the qualitative data show participants describing positive changes, both to their individual health and experiences of their neighbourhood as a result of participation in the *Well London* activities. However, equally significant was the degree of variation in how these changes were perceived between each neighbourhood, which was modulated by: mode of delivery, characteristics of individuals, neighbourhood history and attitudes to social interaction. As a consequence, each area is described separately in the results and the basis for using quotes is to represent what was said in relation to the themes.

Overall, participants identified the importance of social interaction as a crucial component of participation in the *Well London* activities. For example, a social gathering that included eating healthy food; gardening and opportunities to chat over a cup of tea; women feeling safer when sharing an evening walk together. Participation in practical health-related activities was only beneficial within a social context. By comparison, non-participants, despite their individual attempts at improving their own health, experienced no benefits either from the efforts to change eating or physical activity levels, or from being around others in the community who were participating in the *Well London* activities:

> So I was left a very lonely bunny for quite a while. I don't like going out walking all the time on my own, I don't like going swimming on my own. I love to do it, but I don't like doing it on my own. If there was a group going, I would go Mary, age 48, Irish, non-participant

Furthermore, a small number of non-participants felt excluded from the *Well London* interventions, suggesting there may be some non-beneficial effects. For example, in response to a question about positive changes on the estate as a result of *Well London*, one resident commented:

> Well, that "getting better" is a matter of opinion, because I look on it now as a ghetto. It was an unruly estate before. It has quietened down, but now it's a ghetto. I don't go out anymore, I don't do anything anymore. No. No. And I hardly even talk to people now. I mean, I'll sit out at my doorstep and, you know, a lot of people'll stop and chat to me, but I don't really like it. Karen, age 54, White British, non-participant

Despite scoring high on the 'Indices of Deprivation',[29] the three *Well London* intervention neighbourhoods will be described separately in acknowledgement of their diversity and to bring out the nuances of how place impacts participation and any consequent outcomes.

*Hartfield* is a large housing estate built in the 1950s, comprising low-rise blocks constructed around a series of rectangular grassed areas. Although the most homogenous of the three neighbourhoods in terms of population and environment, prior to *Well London* it was socially fragmented with a dearth of community activities.

Preintervention descriptions of Hartfield included:

> And the word 'Hartfield' put horror—it was notorious. Everybody who was difficult was dumped here. Margaret, 59, White British

> It's a dumping ground. It always has been a dumping ground. You know? I begged not to be put on there. I've been there 21 years. Liz, 48, Irish

> I will tell you straightaway there was no life in the community before the arrival of Well London. No, that is the summary of the whole thing; where you are living in an area where there was no life. Clifford, 46, African (Uganda)

Postintervention, Hartfield respondents reported the most substantive change in experienced health benefits of all three neighbourhoods. Factors that facilitated this included: (1) a proactive, charismatic *Well London* coordinator; (2) increased safety following changes in policing methods on the estate, instigated by the coordinator; (3) a high number of proactive volunteers; and (4) residents as stakeholders through the estate's Residents Committee, set up by *Well London*. Benefits described included: enhanced feeling of social cohesion, new knowledge about health, involvement in estate-wide activities, improved relations with neighbours, less complaints about the neighbourhood's lack of cleanliness, safety and violence (see tables 2–4; social interaction).

*Eastford* has undergone extensive regeneration over the past decade, including funding to develop community projects that promote health. This had generated an ethos of community participation and differentiating the *Well London* interventions from these other activities was subsequently more difficult, especially its effects on mental health and well-being. Despite this, the positive changes experienced here by participants refer specifically to the *Well London* activities.

Preintervention, Eastford was already defined as a place where things happened:

> Eastford is great—there's so much to do here! Jermina, 26, Bangladeshi.

Postintervention, respondents experienced some change. Benefits included (1) a sense of autonomy from volunteering and involvement in managing and running activities; (2) feeling productive and useful; (3) increased knowledge of food/cooking and improved health; and (4) enhanced feeling of social cohesion.

## MOUNTSIDE

Mountside is a neighbourhood of contradictions, characterised by a geographically dispersed, ethnically and socioeconomically diverse but transitory population and a reverse trend in terms of regeneration:

**Table 2** Results: Hartfield

| Hartfield | Reported benefits in HE, PA, MHWB and SI |
| --- | --- |
| HE | ▶ *So you know, after walk we have this exercise to stretch ourselves, and then after that we used to have fruit. Yeah, so we used to sit in the park and we used to eat fruit and that's how I learn to eat fruit basically.* Priya, 34, Indian<br>▶ *Oh my God, people are healthier now. It's changed, it's completely changed. I say that it's changed because I am involved—I know how much to my own particular health* (and) *the health of my family and how much has changed. I'm able to know more now, I know what to eat, what not to eat.* Thomas, 45, African (Ghana) |
| PA | ▶ *Like before, you know, I used to find walking was kind of one of the painful things, yeah, I wouldn't bother to walk, I would rather take bus rather than walking, but now I feel like, now—rather than taking bus or anything, let's just walk, it's not going to take me that long.* Sandra, 43, African (Uganda)<br>▶ *And we have some people who want to go night walk—like the Somalians. If the place is dark, they would like to walk. Because of, you know, night-time you can also wear your trousers—so that they can walk faster.* Joyce, 38, African (Nigeria) |
| MHWB and SI | ▶ *The fact that it's made me proud of myself and the whole project and the whole community, because it's made people come in to do the activities.* Bernard 42, African (Ivory Coast)<br>▶ *It has made the community come together, that's what I've seen anyway, people have come together, which is very good.* Claudette, 37, African (Sierra Leone)<br>▶ *Yes we used to be on our own, nobody say hello to each other, but because of Cheryl* (*Well London* coordinator), *Well London came to this place* (and) *it start connecting us.* Lorraine, 39, African (Uganda)<br>▶ *Yeah the police, which are responsible to this area, yes its changed a lot because now they can say 'hello' to you Sometimes even the kids, if they see them playing outside they will stand and speak with them, and ask them 'are you with elderly adults or are you alone?' And so forth and we are happy for that.* Margaret, 41, African (Ghana)<br>▶ *I think that it does a good thing—Well London came to help out. We did a food basket with five foods, I did that as well so. I know the women come to do, they have sewing classes, and it's just—but it's for the coming together, the community together, that's what I think.* Clara, 42, Bangladeshi<br>▶ *I've been proud to say that this is one of my proudest periods in regards to this community. Yeah, this is because our efforts that has been put in place by Well London and followed by Well London volunteers. Last year I was just a volunteer to Well London but this year I am the chairman of Hartfield Estate—the estate now compared to what it was in the past, it's a name at least to be proud of.* Frank, 39, African (Nigerian)<br>▶ *Before we started living here I heard* (that) *the estate wasn't really nice, it wasn't really good. Yeah, in terms of gangs and all those things. But I was, well initially I was a bit scared…no this place has changed now. It's not like the way it used to be before* (Well London), *there's a lot of cameras around, and then there's this local police office behind, just around there, which is really good, so.* |

HE, healthy eating; MHWB, mental health and well-being; PA, physical activity; SI, social interaction.

It was a transient population so you'd get people move in for three months, as I say, trash the place or do whatever. Paula, 45, White British

It became what I can only describe as a dumping ground for literally anybody. There was no perspective on who was living where and next to whom; people were just thrown into the flats regardless of background, criminal intention or anything. Mohan, 52, Indian

When we first moved here it was gorgeous. Oh, you couldn't have wished for a more idyllic place to live. It was quiet, it was flowers, it was lovely neighbours. But of course, a lot of our neighbours then had been here since the block went first up in the '60 s, so they were all getting old and consequently all started to die and then their families sold the flats to housing associations. And you go from there. Monica, 62, White British

The loss of facilities such as a local cinema, shops and other community activities resulted in the main street consisting of fast food outlets and budget shops; high

street brands that used to exist had moved away, apart from a large supermarket. There were pockets of privately owned terraced housing divided from local authority tower blocks, marking a clear socioeconomic boundary. Attitudes to *Well London* were similarly divided; some viewed the interventions positively, and some not.

Postintervention, Mountside respondents recounted little change; positive change was commented on only in relation to the mental health and well-being activities. Factors that prevented change were: (1) lack of effective, coordinated local leadership; (2) dispersal and transience of the local population; (3) lack of cohesive environmental planning; and (4) strong sense of neglect and 'being forgotten' by residents.

## CONCLUSION

Participants described an overall positive impact from the *Well London* project activities, but the data also reveal

**Table 3** Results: Eastford

| Eastford | Reported benefits in HE, PA and MHWB and SI |
|---|---|
| HE | ▶ Yeah, whereas say somebody comes in on Tuesday and does a little bit of cooking—it's quite quick, but with the 'Cook & Eat' it was more in depth and they explained things better and you could ask questions and all things like that, yeah, it was much better. *Clare, 38, White British* |
| | ▶ *Earlier I used to be like, junkie foods eating; crisps and all those things. Now it's like more fruit and vegetables and salad in my diet.* Shubha, 28, Indian |
| PA | ▶ *It was great, it's fantastic—I cannot express how good it is to get in there and get your hands dirty, and to see everybody else doing the same thing.* Sarah, 34, White British |
| | ▶ *Yes, I do a lot, because I'm doing them exercises it's helped me, it's good for my health, I feel much better, I can breathe properly. And you make friend. Yeah, it's good for me—I go out, and you meet friends.* Tricia, 72, Caribbean |
| MHWB and SI | ▶ *I feel I can keep my mind going and I feel like my mind has to be active because I don't want to sit down and get depressed or something. If I think bad things then I won't be doing nothing and I don't want to go like that yeah.* Maureen, 48, Irish |
| | ▶ *I feel so much more confident that we can make this move on; the thing we were given was confidence building. I think that sort of confidence building was something I didn't see—yeah, running an organisation, running that level of budgeting and planning.* Michael, 50, White British |
| | ▶ *You can see it, just a healthier lifestyle: people busy all the time, people—not so much arguments and you see that less and people are a lot more sociable as well.* Pat, 36, Black British |

HE, healthy eating; MHWB, mental health and well-being; PA, physical activity; SI, social interaction.

a complex and nuanced picture of if and how outcomes were achieved with two key findings. First, it shows how neighbourhood-level changes did not lead to benefits among those who did not participate in project activities. Second, the characteristics of neighbourhoods, both social and physical, were fundamental in moderating whether people participated, the nature and extent of the consequent benefits, and any reported changes in health practices. Therefore, participation is dependent on the provision of particular elements that support it;

**Table 4** Results: Mountside

| Mountside | Reported benefits in HE, PA and MHWB |
|---|---|
| HE | ▶ *Yeah, you know children like chips, sausages, yeah. Just sometimes I'm cooking chips—every time Turkish foods; rice yeah. You know, my older one all the time she wants outside, McDonalds, chicken, chips, she's eating too much. And everywhere this food. I'm telling her 'you know, too much oily inside, you no eat' and she's not listening to me.* Hanife, 36, Turkish |
| | ▶ *You can see the higher fast food intake, zero exercise, high alcohol and stressful kind of lifestyles that people lead. And this is also supported by the number of fast food outlets that thrive in these areas.* Mohan, 31, Asian British |
| PA | ▶ *We've got one park over the road but, again that's a dangerous place. We've had murders over there, we've had people killing the swans to eat and people sleeping rough over there. So of course, parents weren't taking their kids over to the park, and you can't blame them, I wouldn't go over there.* Marie, 46, White British |
| MHWB and SI | ▶ *It was fantastic, it brought up a lot of issues and a lot of practices and things that I'm already aware of, and I really, really enjoyed it'.* Saroja, 27, Asian British |
| | ▶ *Oh yeah, and I wish it could continue, I really, really do, because I think it's started to actually break down a few barriers. We were all really sad when it ended and I thought; this is something that could really build up. And I just wish we could have* Well London *permanently. It was a really nice thing, and because it came to this area it made us think, well we are important, it's come here. I know it came here because we were a deprived area but people are listening to us. They're trying to do something to help us. And like I say, the worst thing is that we haven't got it (*now*). If you can bring it back I'd be ever so grateful and so would a lot of other people.* Karen, 41, White British |
| | ▶ *I came away having learnt a lot more about the other women—appreciating them more, yeah, I think that's word should be put in there; appreciating other people, not just cultures but people themselves.* Molly, 45, Caribbean |
| | ▶ *It takes the form of exercise when I can be bothered. I will say I'm a bit lazy sometimes, so you'll do it and then it's like you don't want to take it on, on your own, so you do need motivation* Jan, age 36, White British |

HE, healthy eating; MHWB, mental health and well-being; PA, physical activity; SI, social interaction.

namely a socially cohesive environment in which to get to know neighbours; a safe environment that is well maintained; access to affordable, nutritious food; a degree of autonomy that allows residents to be involved in decision-making and thereby improve confidence and self-esteem. These findings are substantiated through the statements of participants and shown throughout, in comments concerning the importance of friendships made, improvements to eating habits, and increased feelings of safety, post *Well London*.

The role of the *Well London* coordinators also emerged as an important theme across the three areas. With their active involvement through coordinated organisation of volunteers, a commitment to the area shown by their understanding of local issues, participation was more successfully implemented. In Hartfield, for example, residents frequently cited their coordinator as a 'boundary crosser'[30]; she was pivotal in encouraging and facilitating their involvement in activities. By contrast, coordination of the activities in Mountside was deemed problematic by many, apart from those attending DIY Happiness groups, which were identified as positive because they acknowledged residents' sense of deprivation. Activities that did less well were those deemed to be out of touch with local needs, that is, 'fun' activities were less successful than those seen to have direct relevance and benefit, such as stress management. In Eastford, residents were encouraged to lead projects and be involved in decision-making and subsequently, external *Well London* leadership and coordination was mentioned less, whereas the benefits of taking a leadership role were spoken of frequently.

Well-being was a central requirement for the exercise of personal agency, which in turn enabled participation in the *Well London* activities. Well-being was tied to factors such as being able to live in a socially cohesive and safe neighbourhood where neighbours respected one another and where problems were recognised and acted upon by local authorities. Once engaged, there was an apparent feedback loop whereby further enhancement of well-being increased personal agency and lead to increased involvement in the activities which then lead to changes in attitudes and practices in eating, exercise and mental health. Participants' well-being, agency and participation also interacted with their sense of place, again in an iterative fashion. Following improvements to the physical environment, such as direct involvement of local police in providing safer spaces, further enhancements of well-being and agency were described. Well-being in this instance, appears to be a crucial mediator between agency, participation and improved health practices. A recent review of individual experiences of community engagement also found that active participation in community initiatives has important psychosocial benefits for participants that include enhanced feelings of personal confidence and self-esteem, as well as enhanced social relationships and social cohesion within a community.[31 32]

In this study, participation was not a simple binary variable and quantitative measures alone did not pick up the subtleties and complex variations. Our findings show that participation is a complex and dynamic process with well-being at its core, acting as a catalyst that enables participation through a related sense of personal agency. Through further enhancement of well-being and associated social cohesion, improvements in health practices were experienced, just as in its absence no benefits were recounted. However, participation was not universally desired; some reported feeling excluded from the *Well London* interventions and the subsequent changes taking place in the neighbourhood (see participant comment, p.4), while others reported little interest in taking part because improving health was neither a priority nor an personal goal.

These findings also confirm that health practices are not a separate 'capsule' of behaviour,[33 34] but are embedded within particular social, cultural and physical milieus. Across the three neighbourhoods however, there was a clear gradient of change with the greatest change seen in the presence of: (1) involvement and support of external organisations; (2) personal and collective agency enhanced by effective leadership in project activities; (3) social cohesion fostered by *Well London* activities. Where the physical and social environment remained unchanged, there was less participation and therefore fewer benefits. Also, each area reflected considerable variation in levels of maturity and self-management: each was graded in terms of what progress was possible, Mountside being at the beginning of area-level change with safety and environmental pollution still an issue in contrast to Eastford, which featured a more developed and progressive attitude due to investment, both financial and from the local authority. Hartfield was in the midst of significant degrees of change through necessity and a relatively recent influx of enthusiastic residents supported by an equally enthusiastic coordinator.

As others have identified, the dynamics of participation from the perspective of individual agency have been neglected.[17 18 32] In addressing this, our findings show participation as a dynamic and flexible process with agency at its core. Also, the community engagement approach fed back into and reinforced feelings of well-being and agency and thus encouraged and supported changes in health practices. The findings are consistent with elements of Popay's[17] proposed pathways by which community engagement leads to health outcomes, and specifically that social capital/cohesion and community empowerment are important intermediaries.[17] Additionally they point to the need for further understanding of how these interact with agency, well-being and empowerment at the individual level and in different social contexts, in order to achieve inclusive engagement with such programmes. As Popay argues,[25] people's own ideas need to be incorporated fully in the design and delivery of proposed health interventions.

**Acknowledgements** The authors thank all of the study participants for giving generously of their time; the project delivery partners for their assistance; our

colleagues at the University of East London, University of Westminster and the London School of Hygiene and Tropical Medicine for their support and cooperation.

**Contributors** JD was involved in data collection, interpretation, analysis and writing. SJ and AC took part in data analysis and interpretation. RL was involved in data collection and analysis. MP and GP were involved in data interpretation. AR led the project. AD took part in data interpretation, analysis and writing.

**Funding** This work was supported by the Wellcome Trust (grant code: 083679/Z/07/Z).

**Competing interests** None.

**Ethics approval** Granted by University of Westminster Ethics Committee.

**Provenance and peer review** Not commissioned; externally peer reviewed.

**Data sharing statement** No additional data are available.

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
