## [Reviewer comments · BMJ Open]

Some articles will have been accepted based in part or entirely on reviews undertaken for other BMJ Group journals. These will be reproduced where possible.

ARTICLE DETAILS

TITLE (PROVISIONAL)	'Well London' and the benefits of participation: results of a qualitative study nested in a cluster randomised trial
AUTHORS	Derges, Jane; Clow, Angela; Lynch, Rebecca; Jain, Sumeet; Phillips, Gemma; Petticrew, Mark; Renton, Adrian; Draper, Alizon

VERSION 1 - REVIEW

REVIEWER	Susan B. Rifkin Senior Research Fellow Institute of Social Psychology London School of Economics I have no competing interests.
REVIEW RETURNED	14-Aug-2013

GENERAL COMMENTS	There were no statistics in this paper. There are not supplemental documents. I found this paper well presented and clearly structure. I also found the methodology robust and well described. I do have some concerns. Firstly I found some concluding statements too strong for the evidence. The statement that "Well – being was in this instance a crucial mediator between participation, agency and improved health practices". I would like the authors to moderate this by say "Well-being appears to be". There were only 61 respondents out of a much wider cluster randomized trial. The statement as it is seems to infer this finding can be generalized to the whole study population. Qualitative findings do not allow such generalizations. In the paragraph discussing the reasons for gradient changes across the 3 communities, the discussion does not refer to the evidence about the role of leadership. I would like to see a discussion about leadership in terms of the differences in the communities. Finally, I am concerned the findings don't contribute very much to what we know about community participation and health improvements. They add to the evidence that has already been documented but don't really provide new insights. I would suggest the authors strengthen their conclusion by placing their evidence in the context of analysis of other reviews, two of which are referenced below and to look at the work of Cathy Campbell at the London School of Economics. Her studies with colleagues that can be found on the Internet. Campbell explores social theories in the context of health and participation that might provide a basis for a contribution in terms of a more in depth insight for the information presented in this paper.
--

	References: Preston, R., Waugh,H., Larkins,S. and Taylor,J. Community participation in rural primary health care: intervention or approach? Australian Journal of Primary Health, 6,4- 16, 2010. Liberato, S., Brimblecombe, J., Ritchie, J. Ferguson, M. and Coveny, J. Measuring capacity building in communities: a review of the literature. BMC Public Health 2011: 11:850 http://biomedcentral.com/1471-2458/11/850.
--	--

REVIEWER	Stronks, Karien Academic Medical Centre / University of Amsterdam, Public Health
REVIEW RETURNED	01-Nov-2013

GENERAL COMMENTS	The idea of exploring the mechanisms that generate effects of a community engagement programme (in this case in London) is extremely interesting and relevant, both from a scientific and societal perspective. In addition, the design of this study, a longitudinal qualitative study, embedded in a cluster randomized trial, is interesting and very well chosen. However, the research questions as they have been posed in this manuscript are not clearly defined. Nor are the central terms used in the manuscript. First, it is not clear what the actual outcome measure is. Sometimes the authors use the term engagement, sometimes participation. Which phenomenon are they referring to? Participation in concrete activities? Or participation in the government of the programme, or the development of the interventions? Second, the research questions are very broad, too broad in my opinion for one single paper. The research questions refer to participation as well as change in health (practices) as well as link between environment and attitudes. None of these objectives have been systematically explored in the paper as it stands. Third, the theoretical framework has been described rather loosely only. The authors in the introduction refer to terms as agency, autonomy, social contexts without giving a definition or making the relations between these terms. The results and conclusion as they stand suffer from major flaws:  1. The description of the results is superficial, probably related to the fact that so many research objectives have been included in this paper. E.g. at p. 6, they describe the changes that have been taken place in the eyes of the respondents, without explaining what these changes actually consists of. 2. The authors do not show the underlying data on which their observations are based. They give some excerpts, but these in itself do not always refer to elements of the interventions and the observed changes in outcome measures, which is necessary to answer the central research questions, referring to the mechanisms that account for potential effects. 3. The central conclusions refer to the interaction between participation, agency, well-being etc., and the influence of context. The underlying data, however, as presented in this manuscript, do not support these conclusion, nor do they give sufficient insight into the mechanisms the authors refer to in their conclusions.
---

	4. It seems to me that the overall conclusion that participants describe an overall positive impact of the project activities is not justified by the data. For the authors do not systematically report on these changes. In addition, the type of changes have not been clearly described The topic of this paper is very well chosen, and the study the paper is based upon is extremely interesting. The paper as it stands, however, suffer from major flaws. I would stimulate the authors to write a new manuscript, in which they take into account the methodological flaws in the paper as described in my review.
--	--

VERSION 1 – AUTHOR RESPONSE

1. The description of the results is superficial, probably related to the fact that so many research objectives have been included in this paper. E.g. at p. 6, they describe the changes that have been taken place in the eyes of the respondents, without explaining what these changes actually consists of.

Response:

The research objectives have been changed to reflect greater clarity and focus, as follows:

- 1. To identify how and why individuals participated in Well London activities;*
- 2. To explore the different project components that enabled people to improve their health practices and sense of ‘well-being’;*
- 3. To identify factors in the social and physical environment that influenced attitudes to health.*

Additionally, in the results section on page 6, further details are provided with examples, showing the changes reported by participants. As this is a qualitative paper, participant's statements and observations are central throughout the paper.

2. The authors do not show the underlying data on which their observations are based. They give some excerpts, but these in itself do not always refer to elements of the interventions and the observed changes in outcome measures, which is necessary to answer the central research questions, referring to the mechanisms that account for potential effects.

Response:

The ‘data’ in this instance, is formed from the personal statements taken form in depth interviews with participants. Yes, participants statements do not always reflect elements of the interventions – this is where the complexity lies; responses to the interventions are not necessarily ‘in tune’ with the interventions themselves, hence our conclusions that “*characteristics of neighbourhoods, both social and physical, were fundamental in moderating whether people participated, the nature and extent of the consequent benefits, and any reported changes in health practices*”. We understand that ‘outcome

measures' and 'mechanisms' are central to quantitative studies, but due to its qualitative nature, this study takes a perspective that centres on individual experience which we appreciate is more difficult to measure.

3. *The central conclusions refer to the interaction between participation, agency, well-being etc., and the influence of context. The underlying data, however, as presented in this manuscript, do not support these conclusion, nor do they give sufficient insight into the mechanisms the authors refer to in their conclusions.*

Response:

We have added changes to the conclusions on page 11, line 28. Also, our evidence showing what 'mechanisms' lead to our conclusions, states that participation is dependent on agency and a sense of well-being that is supported by a positive social environment. Participants describe the changes to their social environment as pivotal to their participation and how agency was enabled when supported by the interventions. Those who did not participate, did not show any changes or report any benefits.

The insights offered in this paper conclude that in order for health interventions to succeed, individual agency and well-being needs to be supported and that this is best achieved through improvements to the social environment.

We have also added further details on the role of the Well London coordinators, which we think will help support the arguments being made:

"The role of the Well London co-ordinators also emerged as an important theme across the three areas. With their active involvement through co-ordinated organisation of volunteers, a commitment to the area shown by their understanding of local issues, participation was more successfully implemented. In Hartfield for example, residents frequently cited their coordinator; as a 'boundary crosser'[30], she was pivotal in encouraging and facilitating their involvement in activities. As a she By contrast, co-ordination of the activities in Mountside were deemed problematic by many - apart from those attending DIY Happiness groups, which were identified as positive because they acknowledged residents' sense of deprivation. Activities that did less well were those deemed to be out of touch with local needs i.e 'fun' activities were less successful than those seen to have direct relevance and benefit, such as stress management. In Eastford, residents were encouraged to lead projects and be involved in decision-making and subsequently, external Well London leadership and co-ordination was mentioned less, whereas the benefits of taking a leadership role were spoken of frequently".

4. *It seems to me that the overall conclusion that participants describe an overall positive impact of the project activities is not justified by the data. For the authors do not systematically report on these changes. In addition, the type of changes have not been clearly described (see above).*

Response:

The boxes identify how attitudes among participants have been shaped by participating in Well

London. These report some improvements to health, but centrally, improvements to a sense of well-being were more important and arose through becoming involved in social activities that enabled individuals to exercise individual agency. This has been clarified further on page 11, lines 28-32:

“Therefore, participation is dependent on the provision of

particular elements that support it; namely a socially cohesive environment in which to get to know neighbours; a safe environment that is well-maintained; access to affordable, nutritious food; a degree of autonomy that allows residents to be involved in decision-making and thereby improve confidence and self-esteem”.

And again on pg 11, line 47-50:

“Well-being was a central requirement for the exercise of personal agency, which in turn enabled participation in the Well London activities. Well-being was tied to factors such as being able to live in a socially cohesive and safe neighbourhood where neighbours respected one another and where problems were recognised and acted upon by local authorities”.

VERSION 2 – REVIEW

REVIEWER	Karien Stronks Dept. of Public Health, Academic Medical Center/University of Amsterdam, the Netherlands
REVIEW RETURNED	25-Jan-2014

GENERAL COMMENTS	Thank you for the revised manuscript, which has been improved, especially with respect to the central aims and research questions. I am still concerned, however, about the verifiability of the data. Evidence must be verifiable in order to be scientific. Many of the statements in this manuscript are, however, not verifiable for me as a reader. E.g. the authors state at p. 7: "Factors that facilitated this included: a) a pro-active, charismatic Well-London coordinator; b) increased safety following changes in policing methods on the estate, instigated by the coordinator; c) a high number of proactive volunteers; and d) residents as stakeholders through the estate's Residents Committee, set up by Well London". My question is: how do I know that I would draw the same conclusion, in view of the data used in this study? In other words, how should I verify the validity of the statements? I do of course realise that qualitative studies have a different methodology than quantitative research, but the requirement of verifiability applies to both type of methodologies. In this case, e.g. excerpts of the interviews are necessary to substantiate these statements, as the authors do in the boxes with regard to the reported benefits of the interventions. The same applies to other conclusions as drawn in the paper. E.g. at p. 11 "Therefore, participation is dependent on the provision of particular elements that support it; namely a socially cohesive environment in which to get to know neighbours; a safe environment that is well-maintained; access to affordable, nutritious food; a
--

	degree of autonomy that allows residents to be involved in decision-making and thereby improve confidence and self-esteem." These conclusions do make sense, but the question here is: how are they supported by the data? How, e.g., do the authors come to the conclusion that a safe environment that is well maintained is a prerequisite for people to participate? Additional comment: 'Participant' is now defined in different ways in the manuscript. Whereas in the definition in the introduction, participation refers to the specific activities, in the method section, 'participant' is also defined as residents who volunteered in the delivery of the activities. This seems inconsistent. I hope I have been able to further clarify my concerns about the manuscript. The topic is extremely important, and the methodology of a qualitative study within a quantitative effect study extremely interesting. I gave my comments with the intention to further strengthen the paper.
--	--

VERSION 2 – AUTHOR RESPONSE

Many thanks for the comments and careful consideration of our paper, which is much appreciated. We have taken note of these and made amendments as follows:

Pg 3. We have clarified that 'participation' is inclusive of both participation in the interventions and in volunteering

Pg 8. We have provided additional clarifications both in reference to the Well London co-ordinator and added an extra quote to highlight the changes in relation to both proactive volunteering, the changed environment and policing, as a consequence of WL

Pg 11. We have drawn attention here to the statements of participants, who refer to changes in social interaction, eating habits and feelings of safety.

VERSION 3 - REVIEW

REVIEWER	Karien Stronks Dept of Public Health, Academic Medical Center/University of Amsterdam, the Netherlands
REVIEW RETURNED	02-Mar-2014

- The reviewer completed the checklist but made no further comments.